# Physical Blending of Fractionated Bambangan Kernel Fat Stearin and Palm Oil Mid-Fraction to Formulate Cocoa Butter Equivalent

**DOI:** 10.3390/foods12091744

**Published:** 2023-04-23

**Authors:** Norazlina Mohammad Ridwan, Hasmadi Mamat, Md Jahurul Haque Akanda

**Affiliations:** 1Faculty of Food Science and Nutrition, Universiti Malaysia Sabah, Kota Kinabalu 884000, Sabah, Malaysia; 2Department of Agriculture, School of Agriculture, University of Arkansas, 1200 North University, M/S 4913, Pine Bluff, AR 71601, USA

**Keywords:** bambangan, cocoa butter equivalent, fat modification, fatty acid composition

## Abstract

In this study, the physicochemical properties, composition, thermal properties, and crystal microstructure of fractionated bambangan kernel fat stearin and palm oil mid-fraction blends were investigated with respect to a potential cocoa butter equivalent. The blends were prepared in five ratios, and all of the blends exhibited similar physicochemical properties to cocoa butter. Although all of the blends had similar physicochemical properties, the blend containing 70% bambangan kernel fat stearin and 30% palm oil mid-fraction showed remarkable similarity to cocoa butter. The blend had similar fatty acid and triacylglycerol content to cocoa butter with 18.74% palmitic acids, 38.26% stearic acids, 34.05% oleic acids, 15.20% 1,3-dipalmitoyl-2-oleoyl glycerol, and 29.74% 1,3-disteroyl-2-oleoyl glycerol with improved thermostability (high solid fat content at 30 °C but reaching 0% at 40 °C). It also exhibited spherulite crystals with a needle-like crystal structure of 50 µm. This mixture showed good compatibility with cocoa butter at all mixing ratios; hence, it is suggested as a potential cocoa butter equivalent.

## 1. Introduction

Cocoa butter (CB) is an essential ingredient in chocolate manufacturing. CB consists of 25.69% to 25.80% palmitic acids, 36.15% to 36.50% stearic acids, 33.24% to 33.50% oleic acids, 13.80% to 18.10% 1,3-dipalmitoyl-2-oleoyl-glycerol (POP), 26.30% to 40.20% 1-palmitoyl-2-oleoyl-3-stearoyl-glycerol (POS), and 19.20% to 29.70% 1,3-stearoyl-2-oleoyl-glycerol (SOS) [1,2,3,4]. These compositions affect physical properties such as solid fat content (SFC) and thermodynamics, resulting in polymorphic transformations and temperatures that cause chocolate to solidify at 20 °C to 25 °C and then melt completely at body temperature [5]. The continuous fat phase of CB is also responsible for the texture, hardness, melting properties, flavour, desirable *β*-polymorphism, and cooling or smoothing effect when the product is consumed [6,7]. This behaviour corresponds to the V(*β*)-polymorph, which provides the optimal melting behaviour [8]. This fat can also influence the quality parameters of chocolate, such as the snap, demoulding, colour, and gloss [4].

CB is expensive because of the tight supply and declining production that exceeds the global demand. It is very valuable and more expensive than other vegetable fats and oils [9]. Therefore, the industry is keen to find cost-effective and compatible alternatives to CB. Various vegetable fats can be used as blending ingredients to produce CB alternatives [10]. A CB equivalent (CBE) with similar properties to CB can be developed from these fats via physical blending [11]. Physical blending produces a blend that is compatible with CB, which can be incorporated wholly or partially into CB. This CBE can be mixed in any ratio due to its similarities without changing the physical, rheological, and melting properties of CB. This behaviour can also be achieved by preparing a CBE with similar crystallisation properties [8]. Conventionally, CBE is prepared by blending high-POP-content fat, such as palm oil mid-fraction (POMF), with SOS-rich fats, such as mango kernel fat and high-oleic high-stearic sunflower stearin [8,12,13]. POMF with an iodine value (IV) of 45 g iodine/g produced via two-stage fractionation has also been reported as a blending ingredient with other vegetable fats to formulate CBE [13,14,15]. POMF has a high content of POP, with values of 51.80% (51.60% palmitic and 35.6% oleic acids), resulting in a steep SFC [10,16,17].

Compared to other vegetable fats, POMF is economical and suitable as an ingredient for CBE production, due to its availability, cost, and composition. Given its unique melting and SFC properties, POMF has a wide range of applications and is suitable for confectionery fillings and compound chocolate [10]. The high POP content in POMF makes it desirable to be blended with POS- or SOS-rich fat sources to resemble the triglycerides (TAG) composition of CB [4]. In contrast, the second stearin (BKF-SS) of bambangan kernel fat (BKF) is fractionated from the crude BKF via two-stage fractionation that has 64.7% of SOS-TAG [18,19]. The symmetrical SOS-TAG in this fraction can be used as a blending component with palm fraction for CBE production [16,20]. Currently, bambangan trees are being planted in backyards or orchards in response to the increasing demand for this fruit; thus, the cultivation of bambangan fruit in Sabah shows a steady growth from 121.6 to 133.03 and 121.6 to 133.03 tonnes from 2013 to 2015 and 2016 to 2020, respectively [3,21].

Therefore, the fractionated POMF and BKF-SS were physically blended in a different ratio to produce CBE with similar properties to CB. The physicochemical properties (i.e., IV, slip melting point [SMP], acid value [AV], and free fatty acid [FFA]), fatty acid composition, content of TAG, thermal properties, and crystal microstructure of the blends were investigated for their applications as CBEs. Thus, this aim of this study was to formulate CBE and imitate CB in terms of its chemical composition, thermodynamics, and morphological properties via physical blending.

## 2. Materials and Methods

### 2.1. Materials

Bambangan kernel fat stearin (BKF-SS) was produced via multi-stage fractionation [18]. Palm oil mid-fraction (IV: 45 g iodine/g) and CB (IV: 34 g iodine/g) were provided by Sime Darby Malaysia (Selangor) and Malaysian Cocoa Board (KKIP). Fatty acid and TAG standard, as well as analytical chemicals such as sodium thiosulphate, Wijs solution, cyclohexane, hexane, n-hexane, methanol, potassium hydroxide, acetonitrile, and acetone (analytical and HPLC grade), were provided by Sigma Aldrich, Chemie, Steinheim, Germany. The chemicals were of the highest purity.

### 2.2. Preparation of CBE via Blending BKF-SS and POMF

CBE blends were produced by mixing BKF-SS and POMF at different ratios according to the method described by Jahurul et al. [3] and Sonwai et al. [4] with slight modifications. Five blends were produced with a 5% increment of POMF into BKF-SS, as shown in Table 1. The mixtures (50 g) were melted (60 °C) in a hot plate, stirred using a magnetic stirrer (200 rpm for 10 min; SP131320-33-V, Thermo Scientific, Shanghai, China) to ensure homogenisation, and then kept at 4 °C for further analysis.

### 2.3. Determination of Physicochemical Properties of BKF-SS: POMF Blends

The IV (1b-87), SMP (Cc 3b-92), AV (Cd 3a-63), and FFA (Ca 5a-40) for the blends were determined following the recommended official methods of AOCS [22].

### 2.4. Determination of Fatty Acid Profiles

A method described by Norazlina et al. [18] using a gas chromatography-flame ionisation detector (GC-FID 6890N, Agilent, Wilmington, DE, USA) was used to determine the fatty acid composition of the formulated fat samples. First, the fatty acid for the blends was transformed into fatty acid methyl ester (FAMEs) before being injected into the BPX70 column (30 m × 0.25 μm × ID 0.25, SGE, Courtaboeuf, France). Approximately 0.5 g of the melted fat blends was mixed with 2.5 mL of n-hexane and 0.5 mL of methanolic potassium hydroxide (2N). Then, the extracted upper layer of FAMEs was used for the fatty acid determination. The fatty acid composition for the blended fat was identified based on the elution of the FAME standard and reported as % concentration. Analysis was performed using the following condition: oven temperature was set at 90 °C (held for 5 min) and was then increased to 185 °C (8 °C/min; held for 1 min), 200 °C (at 8 °C/min), and to the final temperature of 250 °C (2 °C/min; held for 5 min). The injector (split mode, 1:20) and detector were set at 250 °C, and the helium gas was set at 1 mL/min.

### 2.5. Determination of TAG Profiles

The TAG content for the formulated blends was determined in accordance with the method described by Sonwai et al. [4] with slight modification. A 10% sample solution was prepared by diluting 1 mL of melted fat into 10 mL of mobile phase solution. Then, 2 mL of the 10% fat solution was filtered using a PTFE syringe filter (0.45 µm, Agilent, Wilmington, DE, USA) and transferred into a 2 mL high-performance liquid chromatography (HPLC) vial. Using the following conditions, 10 µL of the sample solution was injected into the HPLC (1200, Agilent, Mississauga, ON, Canada) with a C18 column (Zorbax, 5 L, 4.6 mm i.d. × 250 mm, Agilent, Mississauga, ON, Canada): mobile phase acetone/acetonitrile (70%:30%, *v*/*v*), flow rate 0.8 mL/min, column temperature of 30 °C, pressure 8 to 9 mPa, and detector (RID) temperature of 40 °C. The analysis was run for 60 min. The TAG composition of the samples was reported in % concentration and identified by comparing the retention time of the chromatogram with the lipid standard: TAG mixtures and pure CB. Peaks were identified by comparing peak retention times in the TAG profiles of known CB and TAG mixture standards. The peak area of the chromatogram was used to calculate the percentage of TAG.

### 2.6. Melting Behaviour

In accordance with the AOCS [22] recommended practice for Cj 1-94, melting properties for the blends were assessed using differential scanning calorimetry (DSC Diamond, Perkin Elmer, Waltham, MA, USA). For sample preparation, fat samples were melted at 80 °C for 30 min to ensure that the fat samples melted completely. Next, 3 to 5 mg of fat samples was transferred to the DSC pan (Aluminium volatile pans; 0219-0062, Perkin Elmer, USA) using a syringe (Trumo, 1 mL) with a needle. The weight of the samples was recorded. The samples were then sealed, put into vials, melted for 30 min at 80 °C, and then incubated for seven days at 26 °C to stabilise and temper the fat. The stabilisation of the fat is needed because the TAG can exist in a few crystalline forms such as (I) γ, (II) α, (III) β′_2_, (IV) β′_1_, (V) β2, and (VI) β_1_ while in a state of solid [23]. The less stable forms may change into more stable ones after being stored for a long enough time. A sealed empty pan was put on the reference cell as a standard, and after seven days, fat samples were transferred to the sample cell for analysis. The following condition was used for thermal analysis: heated to 80 °C at 10 °C/min and held for 2 min. The onset temperature (°C), offset temperature (°C), maximum temperature (°C), and enthalpy (J/g) for the fat samples were recorded.

### 2.7. SFC Analysis

Approximately 3 mL of melted (100 °C for 15 min) fat samples was transferred into p-NMR tubes and then placed in a water bath at 60 °C for 10 to 15 min and tempered at 0 °C for 90 to 95 min. Afterward, the tempering process was continued by transferring the tubes to a 26 °C bath and keeping them there for 40 h. The tubes were then held at a specific temperature range (10 °C to 45 °C) for 30 min before being measured. Using a pulse-nuclear magnetic resonance spectrometer (Minispec-mq20, BRUKER, Rheinstetten, Germany), changes in the SFC of the formulated CBE blends as a function of temperature between 10 °C and 45 °C were measured using the official method (Cd 16b-93) of AOCS [24]. The SFC of the measured samples was recorded as %.

### 2.8. Crystal Morphology

Approximately 15 µL of melted (at 90 °C) fat samples was analysed using a polarised light microscope (PLM, DM2500P, Leica, Wetzlar, Germany) at room temperature using 40× magnification. Before analysis, the fat samples were chilled for an hour and then kept at 26 °C for two days. After that, the crystalline structure of the blends was observed and recorded as a microphotograph.

### 2.9. Compatibility Analysis

Compatibility analysis was conducted according to the compatibility test described by Sonwai et al. [4] with slight modifications. The studies were performed using the binary iso-solid diagram of SFC at 20 °C. The CBE (B5), which has similar fatty acid profiles, TAG composition, and thermal properties, was blended with CB at five varying ratios (Table 2). Then, only selected essential parameters such as fatty acid, TAG, SFC, and crystal microstructure were analysed, as described in Section 2.4, Section 2.5, Section 2.7, and Section 2.8. Then, the compatibility was evaluated based on the iso-solid diagram.

### 2.10. Statistical Analysis

Data analysis was performed in triplicate and analysed using one-way ANOVA and the Tukey test in SPSS (version 26). The collected data were expressed as mean ± standard deviation, and *p* < 0.05 was considered to be a significant difference.

## 3. Results

### 3.1. Physicochemical Properties of BKF-SS: POMF Blends

Table 3 presents the physicochemical properties of the formulated blends. IV is a practical quality parameter for determining the hardness and degree of unsaturation of fat samples, and a high IV indicates a high content of unsaturated fatty acid in the samples. POMF had the highest IV, followed by B5, B4, B3, B2, and B1 blends. Compared to BKF-SS [18] and the POMF, the blends showed a significant (*p* < 0.05) decrease in the degree of unsaturation after the addition of high palmitic acid POMF. This change is associated with the changes in the composition of the different fat samples (i.e., increased and decreased content of saturated and unsaturated fatty acid such as palmitic acid and oleic acid). B1 had the lowest IV, indicating that this blend is harder than the other blends. The IV also varies depending on the composition of the blends. However, the IV of the fat blends is consistent with the IV of CB (31.3 to 38.4 g iodine/g) [4,25,26,27].

In addition, the SMP varies according to the IV of the blends. In contrast with the IV, where B5 has the highest values, it has low SMP values. By increasing the POMF content in the BKF-SS: POMF blends, the SMP of the blends decreased proportionally. This change is related to the low melting point of the unsaturated fatty acid (i.e., oleic acid) in B5, which increases IV and decreases the SMP of the fat blend. This fatty acid has a low melting point, which decreases the SMP value of the mixture. Compared to bambangan kernel stearin (36.3 °C) [18] and POMF (28.5 °C to 35.7 °C) [4,27,28], the SMP of the blends decreased to body temperature, which is consistent with the SMP of POMF and CB (27.8 to 35.0 °C) [4,24,26].

The AV is used to determine the total acid content of lipids as a function of the fatty acid components that make up the TAG component. In addition, FFA is a lipid pro-oxidant as it is more susceptible to auto-oxidation than esterified fatty acid [29]. Since then, FFA has been used to evaluate industrial and commercial applications. All of the blends showed lower AV and FFA values than the pure BKF [18], ranging from 1.31 to 2.90 mgKOH/g and 0.62% to 1.46% (FFA as oleic), and 0.55% to 1.32% (FFA as palmitic), respectively. The blends had low acidity levels, indicating that they were edible, with FFA values for edible vegetable oils below 5% [30]. Moreover, these values are consistent with the AV (0.42–2.11) and FFA (1.41%) of commercial CB [4,31]. Thus, all of the blends exhibited good quality and stability similar to that of edible fats such as CB.

### 3.2. Fatty Acid and TAG Composition

The fatty acid contents for all binary blends are shown in Table 4. Similarly to CB, all of the blends had three prominent fatty acids, namely, palmitic, stearic, and oleic acids, with a significant (*p* < 0.05) presence of linoleic and arachidic acids. CB is mainly composed of palmitic, stearic, and oleic acids with percentages of 25.8% to 28.1%, 34.5% to 36.5%, and 33.2% to 33.5%, respectively [4,31]. The fatty acid for CB in Table 4 shows similar values to the reported CB. These fatty acids account for more than 90% of the fat composition and contribute to CB becoming solid at 20 °C to 25 °C. Among all of the blends, B5, with 70% of BKF-SS and 30% of POMF, had low stearic and high palmitic and oleic acid content of 38.26%, 18.74%, and 34.05%, respectively. This blend had comparable stearic and oleic acid content compared to CB, but had lower palmitic acid content. However, these results are consistent with the fatty acid content of the formulated CBE (palmitic acid: 16.3% to 25.2%, stearic acid: 21.9% to 37.2%, oleic acid: 29.0% to 39.6%, and linoleic acid: 3.0% to 3.9%) reported by Sonwai et al. [4], Jahurul et al. [12], Kadivar et al. [32], and Bootello et al. [33]. The linoleic and arachidic acid content of B5 are also comparable to that of CB (linoleic acid: 3.56%; arachidic acid: 1.23%) as reported by Norazura et al. [15]. In contrast, the other blends had high stearic and arachidic acid content compared to CB and the formulated CBE, which would lead to incomplete melting at body temperature. Therefore, the fatty acid composition of B5 is closer to that of CB, with a comparable content of stearic and oleic acids and a low content of palmitic acid corresponding to the CBE (sunflower stearin-based) as reported by Bootello et al. [33]. The blends with a similar chemical composition to CB will have similar textural and melting properties.

According to the fatty acid, the blends had three main proportions of TAG, namely, POP, POS, and SOS, which ranged from 6.41% to 15.20%, 10.59% to 11.92%, and 29.74% to 35.83%, respectively. B1 to B4 showed high content of SOS (30.87% to 35.83%), which could lead to an incomplete melting state of fats. In addition, the contents of POP and POS in the blends were low and did not resemble the properties of CB. B5 showed similarity to the content of POP (13.8% to 18.1%) and SOS (19.2% to 29.7%) of CB [1,2,3,4], while none of the other fat blends showed these similarities. Table 4 shows that the compositions of TAG were more than 70% of that of CB in all fat samples, with a prominence of POP, POS, and SOS. POMF and BKF-SS were dominated by POP and SOS, with values of 44.91% and 63.87%, respectively. The proportion of SOS in the fat blends decreased, and the proportion of POS and POP increased with increasing POMF concentration. Despite similar contents of POP and SOS, the POS content in B5 is significantly (*p* < 0.05) lower than the values determined in this study for CB (26.3% to 40.2%). The low POS content of the blends is due to the addition of the POMF, which changed the palmitoyl content in the TAG composition and reduced the POS content [4]. Therefore, B5 can be used to prepare CBE since these results are in agreement with the TAG profile of CBE presented by Norazlina et al. [10]. Moreover, the similarity of the TAG profiles of these CBE will also lead to good compatibility with the CB. A representative of HPLC chromatogram of TAG in CB and B5 is shown in Appendix A.

### 3.3. Melting Properties

The enthalpy of fusion is a measure of the temperature range in which the first crystal of a solid just begins to melt and the last crystal completes its melting [22]. The stabilised individual fats and the formulated CBE blends were sampled by DSC after seven days to record the melting profiles. DSC is the standard method used to determine the thermal behaviour of the fat samples. This is because the functionality of fats in foods depends on their melting behaviour [33]. All fats were analysed for their thermal properties; although all of the blends had comparable SMP values to CB, the actual thermal profiles for the individual fats and the BKF-SS and POMF blends are still unknown. The DSC melting behaviour of CB, BKF-SS, POMF, and BKF-SS: POMF blends is shown in Table 5. It can be seen that the melting behaviour of the fats is consistent with their fatty acid and TAG composition. BKF-SS, POMF, CB, and the blends showed a single endothermic DSC curve, while BKF showed a broad single endothermic DSC curve.

The results obtained in this study for BKF-SS and CB are consistent with the melting point of SOS and POS, which have melting ranges of 19.50 °C to 35.50 °C and 23.50 °C to 43.0 °C, respectively [34,35]. The results are also in agreement with the melting behaviour of the *β*-form (27.4 °C to 35.0 °C), which is related to the dominant TAG (SOS) in the blends [34]. On the other hand, the melting thermogram of the BKF-SS and POMF blends is improved after the blending process. All of the blends exhibited a narrow melting range between the offset and onset temperatures, indicating that the formulated blends have a suitable profile for the chocolate application. According to Kadivar et al. [32], the melting profile of chocolate is characterised by a sharp, narrow melting range that leads to rapid melting at body temperature, imparts a cool sensation, and causes the release of flavour.

Compared to BKF-SS (15.63 °C) and POMF (19.58 °C), B1 to B5 (8.55 °C to 14.71 °C) have a narrow melting range, with B5 being the narrower one (8.55 °C). Thus, B5 melts the fastest, and this profile also illustrated why the blended fat melted quickly compared to the individual fats. The size of the crystal microstructure and the molecular arrangement of the fats also might influence the melting properties. The melting point is higher when molecules are tightly packed together or have a larger size. Overall, all of the blends showed comparable initial and final temperature data from CB (13.00–27.9 °C and 26.8–40.7 °C), indicating their suitability for CBE and chocolate production [11,24,26,31]. Although the blends exhibited comparable melting data to the CB, the formulated CBE blends nevertheless differed in their melting profiles in terms of onset, offset, and maximum temperature. The onset and offset of the formulated blends started at 21.60 °C to 26.71 °C and ended at 35.26 °C to 37.22 °C. These variations can be explained by the different content of TAG (POP, POS, and SOS), which has different melting behaviour and corresponds to the polymorphic form of the TAG.

The integration of POMF in the BKF-SS lowers the content of low-melting TAG (SOO and OOO), which leads to high onset and maximum temperatures in B5. Therefore, it has a narrower melting range and melts faster than the other blends. Although all of the blends have similar melting profiles with CB and are suitable for chocolate fat, B5 shows a faster melting range than the other blends. The melting ranges for B5 are also consistent with *β*_VI,_ the most stable form [35].

### 3.4. SFC

SFC is a key factor in determining the functionality of confectionery fat for CBE application [36]. It measures certain essential sensory properties such as coolness, hardness, waxiness, and heat resistance, using three temperature intervals [37]. The first interval correlates with hardness and brittleness with high SFC values between 15 °C and 25 °C [33]. At room temperature, this phase allows the solidification of CB and gives the chocolate its snap. Heat resistance is then associated with the following interval. Between 25 °C and 35 °C, the fat can survive the melting point with high SFC values. A drastic decrease in SFC values in this interval leads to coolness and creaminess [38], giving the product a cool sensation when consumed.

The waxing phase then occurs between 35 °C and 40 °C; the fat should melt completely or reach 0% at 37 °C, as a trace of SFC amount would cause a waxy sensation. Figure 1 shows the curves of SFC (%) as a function of temperature (started at 10 °C and ended at 45 °C) for the binary blends of BKF-SS and POMF. CB showed the highest SFC at 10 °C, followed by the blends, BKF-SS, POMF, and pure BKF. CB, which was obtained in this study, exhibited all three temperature properties by showing hardness below 22 °C and a gradual decrease after room temperature, reaching 0% at 35 °C. The result is in agreement with that of Sonwai et al. [4], who reported that CB exhibited a rapid SFC decrease as a temperature function, suggesting that it melts completely in the mouth when consumed. It exhibited hardness and brittleness properties indicating that it did not melt at temperatures below 22 °C, and it gradually decreased at temperatures between 20 °C and 30 °C, with less than 10% of SFC at body temperature.

Individual BKF-SS and B5 gradually decreased at 30 °C and reached 0% at 40 °C, moving to a higher temperature than CB. Compared with CB and BKF-SS, the POMF obtained in this study showed a gradually decreasing curve at 20 °C and reached 0% at 30 °C. POMF IV 45 used in this study exhibited softer properties than the hard POMF used in other studies [4,24]. However, blending BKF-SS with POMF significantly changes the SFC profiles of the binary blends, showing a similar SFC to CB. The gradually decreasing curve is similar to that of CB but shows significantly different SFC values. Figure 1 shows that the physically blended fats have higher thermostability properties than CB due to the high stearic acid content, resulting in heat resistance properties. The SFC of the blends at 20 °C (62.92% to 77.36%) is identical to that reported for CB, but the SFC remains high at 30 °C.

Blends with less than 30% POMF (B1 to B4) showed heat-resistant properties with about 2% SFC at 40 °C. The SFC values decreased significantly (*p* < 0.05), with B5 achieving a lower SFC (<15%) at 30 °C to 35 °C and reaching 0% at 40 °C. Similar results were obtained by Jin et al. [24], who blended and interesterified mango seed fat stearin for hard chocolate formulation. Therefore, the appropriate ratio of fat blends of BKF-SS and POMF mixed with CB could lead to similar properties. Among the blends, only B5 did not exhibit a wax phase, indicating the potential of CBE.

### 3.5. Crystal Microstructure

Since the fatty acid and TAG contents of the formulated CBE are comparable to those of CB, the only way to obtain stable symmetrical TAG crystals is to temper the samples with the same parameters as used in the SFC study. The reason for this is that the texture may change under different processing conditions. Based on the microphotographic image, spherulite structures with outwardly branched needle crystals were observed in CB and BKF-SS, while POMF showed continuous granularity. CB (30 µm diameter) showed perfect spherulitic crystals that were disc-shaped. In contrast, BKF-SS (diameter ranged from 30 to 60 µm) and the blends (diameter between 30 and 100 µm) exhibited spherulites with needle-like crystals and granular crystals in the centre. Figure 2 shows the crystalline structures of CB, BKF-SS, and POMF.

The observation is consistent with the compact crystalline structure of BKF-SS (spherulite crystal with needle-like crystal and granular centre) and CB (disc-like crystal with needle-like crystal) reported by Huang et al. [11] and Norazlina et al. [18]. On the other hand, the continuous granular crystal structure of POMF with a diameter less than 10 µm is identical to the microphotograph of POMF as reported by Biswas et al. [17] and Sonwai et al. [4]. From Figure 3, the crystalline structure of BKF-SS: POMF blends changed significantly and showed different crystal morphologies after the blending process. The blends exhibited a large amount of small to large, densely packed crystals that were uniformly distributed with a diameter of 30 to 110 µm. B1 and B3 showed compact crystal structures with diameters of less than 50 µm, while B2, B4, and B5 had diameters of 50, 110, and 50 µm, respectively. These differences are related to the different arrangement of the TAG-FA backbone of the mixtures, which led to the different various textural characteristics [39].

### 3.6. Compatibility between CBE and CB

The compatibility of CBE mixed with CB is shown in an iso-solid diagram (Figure 4). The iso-solid diagram shows the phase behaviour of the diluents and the eutectic/incompatible phase in a fat mixture. The diagram shows that CBE and CB were well mixed, with the line remaining straight across all mixing ratios of CBE and CB. The blends were compatible at all five ratios (80%:20%, 60%:40%, 40%:60%, 20%:80%, and 0%:100%, CBE: CB, *w*/*w*). Accordingly, the SFC measured at 20 °C for the blends lies on a straight line and combines without showing a eutectic phase. This behaviour could be due to the *β*-form resulting from the CBE and CB [40]. Biswas et al. [41] and Jin et al. [25] also reported that ideal or compatible fatty acid mixtures lead to a parallel or straight line, while incompatible mixtures lead to the eutectic phase and the absence of a straight line.

Table 6 shows the fatty acid and TAG content of B5 blended with CB. For all of the blends, the values for palmitic acid (21.42% to 25.20%), stearic acid (32.92% to 36.40%), and oleic acid (32.47% to 36.80%) were closer to the palmitic acid (24.5% to 33.7%), stearic acid (33.3% to 36.5%), and oleic acid (26.3% to 36.5%) of the reported CB [4,15,31]. These fatty acids dominated more than 90% of the composition of the mixture, and the palmitic content in CBE: CB mixtures increased proportionally to the addition of CB. Although the proposed CBE has a lower palmitic content of 18.74%, the CBE did not cause significant changes in the final CB product after blending. These changes are seen in the CBE: CB blends with less than 80% CB; the blends have lower palmitic content, but the value is still closer to that of the CB. The addition of POMF promotes the applicability of BKF-SS, as this CBE has a remarkable similarity in fatty acid composition with CB.

Corresponding to the fatty acid content, the TAG of CBE: CB blends differ slightly from those of CB, but these blends still have values closer to those of CB. The blends that contain <80% of CB have low POS, but the values increase with the proportion of CB, while the content of POP and POS remains comparable. Nevertheless, the TAG profile for the blends is consistent with the CBE prepared from mango seed fat and POMF-base, sunflower stearin and POMF base, and illipe butter and POMF base [1,16,25,26]. The CBE from this study also varied in POS and ranged from 4.6% to 49.53%. Sonwai et al. [4] also reported a low POS (6.7%) content in the CBE in which mango kernel fat was mixed with hard POMF. The CBE obtained in this study has a better TAG profile, in which the POS content is more than 10%, and it shows an increase closer to the CB value when mixed with CB. POMF could stabilise BKF-SS by lowering the SOS content and increasing the palmitoyl content, leading to similar POP and SOS profiles as in the CBE and only slight changes in the POS content, leading to similarities with the above-mentioned CBE.

Figure 5 shows the SFC changes in CB after mixing with CBE prepared from 70% of BKF-SS and 30% of POMF. From the figure, CBE incorporated into CB shows a similar SFC curve and similar trends in the SFC intervals of CB/CBE (15 °C to 25 °C: hardness/brittleness, and 25 °C to 35 °C: coolness/creaminess). The blends exhibit a high SFC in the hardness ranges, while the SFC decreases in the cooling ranges. The results indicate that the blends still exhibit the SFC profiles of CB, and the blend melts completely at 35 °C. Minimal changes are seen in the SFC profiles of CB, especially in the temperature interval. The changes occur only in the SFC values at 10 °C to 25 °C (hardness/brittleness interval). B6 and B7 show significantly (*p* < 0.05) low SFC values at 10 °C (64.6% to 78.11%) to 25 °C (40.44% to 54.48%). The SFC values increase proportionally to the increase in the CB ratio.

This change is related to the changes in the TAG profiles. Although CBE has POP and SOS of CB, the SOO and POS content in B6 are slightly higher and lower, respectively, than in B10 (0% B5, 100% CB). Thus, they have a lower SFC than the CB. The SOO content decreases, and POS resembles the profile of CB with an increasing CB ratio. In addition, the values for all of the blends approach the SFC of CB at 25 °C and melt completely at 35 °C. Generally, all CBE: CB mixtures showed the same trends as CB which showed a gradual decrease after reaching 25 °C. It melts completely at body temperature, and the blends have good compatibility to be mixed with CB in any ratio. The figure shows that CBE1 has exceptional SFC profiles and does not cause waxiness when mixed with CB.

Similar results were obtained when CBE was prepared by mixing mango seed fat with hard POMF [4]. The SFC values for the reported CBE and CBE: CB blends at 10 °C are <60% and <70%, respectively. Therefore, B5 can be used as a CBE and blended in any ratio without the final product having a eutectic affect that causes a soft/waxy feel.

Based on the microphotograph in Figure 6, B7 showed a larger crystal microstructure, followed by B8, B6, B10, and B9. According to their TAG profiles, B7 and B8 have higher SOS content than the other mixtures, which explains the diameter of the crystals. A significant amount of diacylglycerol and triacylglycerols in the blends also contributed to the changes in this profile. Higher temperatures are needed to ensure a smooth feel and complete melt state when these fats are consumed. The SFC values at 30 °C are also slightly higher than the commercial CB, resulting in >1% of SFC at 35 °C and reaching 0% at 37 °C. In order for these blends to taste smooth and completely melt when consumed, higher temperatures are needed. B9, on the other hand, showed the closest similarity to the crystal size of CB, in that it had small crystals of 20 to 30 µm.

The observation showed that B9 (20% CBE1 and 80% of CB) was consistent with the TAG properties of the mixture. This mixture mimicked the TAG content to resemble the phase behaviour of CB with small disc-shaped needle-like crystals. The small crystals produce a similar cool feel and creamy properties (smooth feel) in the end product. The result is also consistent with the SFC profiles of the blends at 10 °C to 15 °C, which have lower SFC values than CB. In general, all CBE: CB blends showed good compatibility with CB by forming similar crystals. Therefore, B5 is proposed as a potential CBE because it can be blended with CB in any ratio and the properties are comparable to the reported CBE, as mentioned in the physicochemical, thermal, and morphological properties of BKF-SS: POMF blends.

## 4. Discussion

Similar to CB, the mixtures are dominated by saturated fatty acids such as palmitic and stearic acid. These fatty acids have become the most common fatty acids in many diets [42]. They may act as coactivators of the heterodimeric nuclear receptor Live X receptor/retinoic acid X receptor (LXR/RXR), which promotes very-low-density lipoprotein (VLDL) production in the liver [43]. Palmitic acid has also been found to cause nuclear translocation and activation of the archetypal inflammatory transcription factor (NF-κB) in cell culture models [44]. Thus, the mixture has nutritional aspects as a potential CBE for health.

The fatty acid content of the blends, especially the saturated fatty acid content, showed significant (*p* < 0.05) changes after the blending process. POMF shows high palmitic and oleic acids due to its high POP content. As POMF content increased, palmitic acid increased in the blends, causing a significant decrease in stearic acid content. These changes affected the physicochemical properties of the fat blends, such as IV and their melting properties. BKF-SS has lower palmitic acid content [18]; therefore, the addition of POMF to the blends improved the stearin properties, resulting in properties comparable to CB/CBE. Increasing the POMF content in the binary blends also led to an increase in the oleic acid composition. This change is related to the high oleic acid content in the POMF.

Statistical analysis of the physicochemical properties, fatty acid content, composition of TAG, and morphological behaviour of BKF-SS: POMF blends showed that the composition of the blends improved toward a similar composition as CB, but showed significant (*p* < 0.05) differences in their SFC properties. The stearic acid content in this CBE is also the lowest among the blends. B5 has similar POP and SOS content to CB and similar TAG profiles to the reported CBE [4,33]. The blends with more than 30% POMF showed high SFC properties and reached only 0% at 45 °C. However, B5, the potential CBE recommended in this study, melted completely at 40 °C and decreased gradually from 30 °C. The mixture also showed melting ranges of CB. The crystal morphology is also consistent with the TAG and SFC profiles of blend B5.

The analysis of crystal morphology is important for determining the consistency of final products [45]. In this study, the crystal microstructures were analysed for morphology analysis to evaluate compatibility with CB. The changes in these profiles correspond to the textural product properties of fat [39]. Instead of polymorphism, the crystal microstructure provides effective results because the similar TAG type in the formulated mixtures leads to a comparable polymorphic form. A small crystal is desirable because it has a larger surface area and can hold a large amount of liquid oil with exceptional creamy properties [46], resulting in a smooth feel when the product is consumed. Small crystals also reduce the hardness of fat, which explains the high unsaturation value in POMF; however, crystals larger than 30 µm in diameter can cause an unpleasant taste sensation.

Comparing the size of other blends, B1 has a desirable crystal structure with a diameter of no more than 30 µm, since a small crystal size leads to the smooth texture of a plastic fat. However, it has a different structure to CB. The spherulitic shapes are barely visible, and the crystals are densely packed with scattered needle-like crystals. CB is characterised by spherulitic and needle-like crystals branching outward, and the blends resemble this structure as the POMF increases. The granular centre also decreased with the POMF content, with B5 having the lowest, resulting in dominant spherulites and needle-like crystal networks. The blends showed large microstructures with a granular centre surrounded by needle-like crystals. This behaviour can be attributed to the phase transition from the *β′* form to the *β* polymorphic form [47].

According to Bootello et al. [33], CB transformed from *β′* to *β* polymorphic form, and these morphological features also exist in CBE. This observation also suggests that the mixtures resemble the morphological features of CB and CBE. Bootello et al. [33] also showed that the crystal microstructure of commercial CBE has a similar large crystal structure. The CBE exhibited a compact, large crystal microstructure; therefore, B5 could have similar textural properties to the commercial CBE. The crystal size of all of the blends is also consistent with the diameter of CB (0.5 to 250 µm), which was determined in a previous study by Asep et al. [48] and Lieb et al. [49]. Therefore, the mixture could have similar textural properties to CB and could be mixed with CB to determine its compatibility state. After blending with CB, the fatty acid profiles varied, but the values were much closer to the desirable range of palmitic acid (24.5% to 33.7%), stearic acid (33.3% to 40.2%), and oleic acid (26.3% to 36.5%) [4,15,31,50].

Compared to the binary blend (B5), the palmitic content increased in the ternary blends (CBE: CB), resulting in minimal changes in the fat composition. B6 (80% of B5 and 20% of CB) had high oleic acid content, which may be attributed to the high oleic content in B5. The linoleic acid content in the ternary blends was also significantly higher than in CB. However, the saturated fatty acid increased with the CB ratio, causing the oleic acid content to decrease. The changes in the fatty acid are caused by the saturated fatty acid content in the CB. The changes in the fatty acid can also cause significant changes in the TAG content of the blend. In relation to the fatty acid content, the arrangement of the glycerol content in the backbone may change, resulting in variations in the TAG profiles. Similar to the palmitic and stearic acid contents, the palmitoyl and stearoyl-TAG content in the ternary blends also showed slight changes.

POS values increased with the increase in CB ratio, while SOS content showed a reverse trend. The changes are consistent with the changes in the fatty acid content. The low-melting TAG content, such as SOO, also showed a decrease proportional to the increase in CB. Blends containing no more than 40% of B5 in CBE: CB blends showed a similar resemblance to the TAG profiles of CB [1,2,3,4]. Therefore, the incorporation of less than 40% B5 in CB could result in similar physical and textural properties of the final product. Although obvious changes can be seen in the profiles of TAG, the values are consistent with the above CBE values; therefore, relevant results were obtained in this study. Although the blends showed good compatibility with CB by showing similar SFC trends to the CBE and CB, the SFC value in B6 and B7 decreased more than in the binary blended CBE.

The high content of low-melting TAG in B6 and B7 presumably caused these changes. The SFC value for CBE: CB blends can also be improved by blending no more than 50% B5 in CB to obtain a final product closely resembling CB in terms of hardness. The SFC for B8 to B10 is consistent with the SFC profiles of reported CB [4,24]. The SFC values for the ternary blends (B8 and B9) are comparable to B10 (100% CB). A proportion of 20% to 40% of CBE from BKF-SS- and POMF-based blends is suggested as an appropriate ratio for blending with CB to obtain a similar final product. Lastly, the compatibility test showed that the SFC and TAG profiles of B5 blended with CB varied considerably, allowing for the possibility that the two specialty fats have different crystal microstructures.

Compared to the binary blend, the ternary blend of proposed CBE and CB showed a smaller crystal size, suggesting that this blend has better creamy properties (smooth) than the binary blends (B5) alone. The diameter of the crystals is also consistent with the reported range of CB (0.5 to 250 µm). The results indicate that all of the mixtures have similar properties to those reported in CB. In response to the TAG and SFC profiles, the blend crystals showed identical structure to the CB with the increased CB content. B10 (CBE1 0% and 100% CB) showed disc-shaped crystals (30 µm) with needle-like crystals radiating outward. Therefore, modification methods such as fractionation and blending processes help in the production of fat blends that have similar properties to CB. These methods are also preferred in the development of alternative fats to replace CB [51]. Calliauw et al. [52] also applied palm kernel oil fractionation to produce CB-substitute fats that can be used directly.

## 5. Conclusions

This study is the first to report on the properties of fractionated BKF-SS and POMF blends. With the right amount of POMF, fractionated BKF-SS blends prepared from BKF could be an alternative source of CBE. The addition of POMF to BKF-SS improved its properties and applicability and showed similarities with CB. Although all BKF-SS: POMF blends had similar physicochemical properties and melting characteristics to CB, only the blend containing 70% BKF-SS and 30% POMF was recommended as a potential CBE. This CBE had comparable fatty acid content and POP and SOS profiles to CB with improved thermostability. This blend also mimicked the spherulitic crystal structure of CB with needle-like crystals. In addition, the CBE showed good compatibility with CB and did not cause significant changes in the crystalline type of CB; it is therefore suggested to have similar textural properties. Therefore, B5 is recommended as a CBE with high thermostability for specific fat applications such as chocolate production. Furthermore, this result suggests that BKF can be used as a novel source of specialty fats to provide alternatives for confectionery.

## Figures and Tables

**Figure 1 foods-12-01744-f001:**
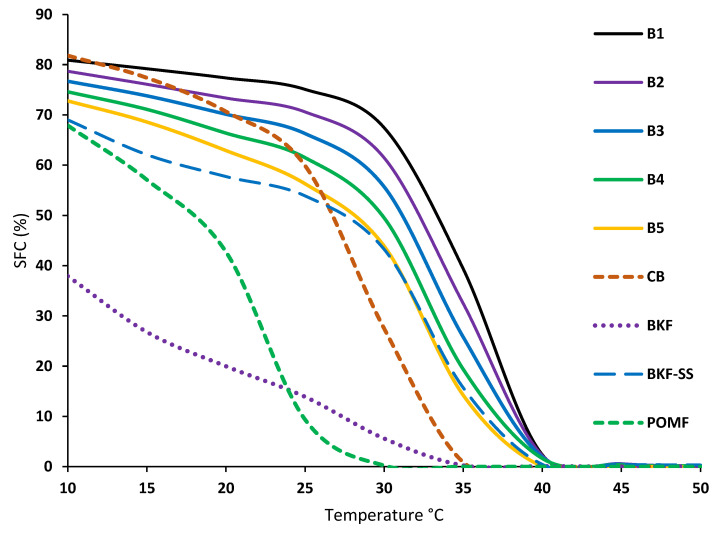
Solid fat content (%) of BKF, BKF-SS, POMF, CB and BKF-SS and POMF blends.

**Figure 2 foods-12-01744-f002:**
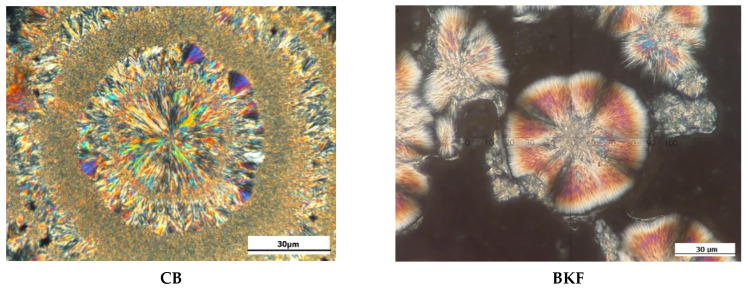
Crystal structures of CB, BKF, BKF-SS, and POMF.

**Figure 3 foods-12-01744-f003:**
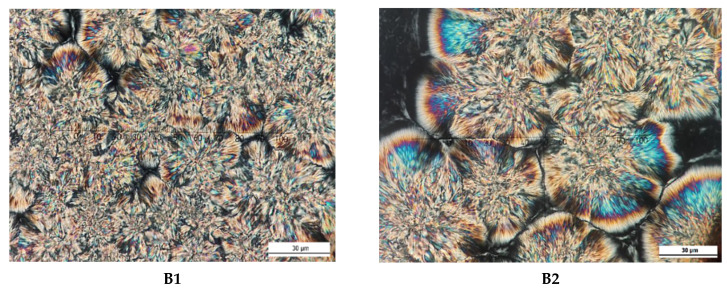
Crystal structure of B1 (90% of BKF-SS and 10% of POMF), B2 (85% of BKF-SS and 15% of POMF), B3 (80% of BKF-SS and 20% of POMF), B4 (75% of BKF-SS and 25% of POMF) and B5 (70% of BKF-SS and 30% of POMF).

**Figure 4 foods-12-01744-f004:**
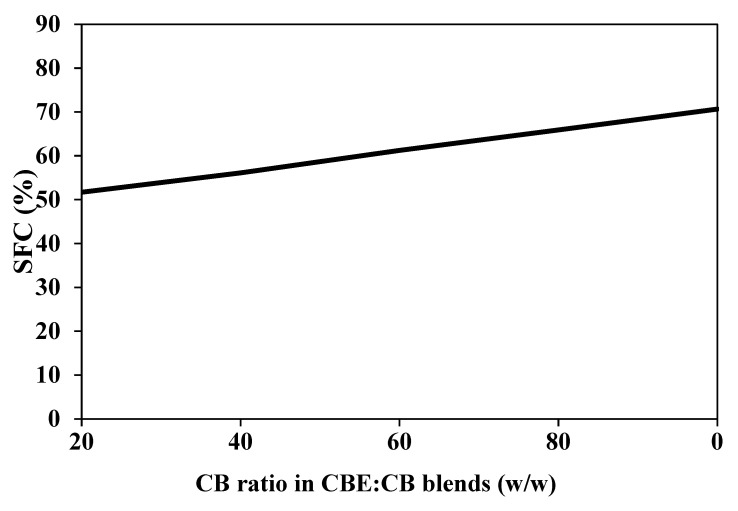
The binary iso-solid diagram for compatibility of CBE with CB, measured at 20 °C.

**Figure 5 foods-12-01744-f005:**
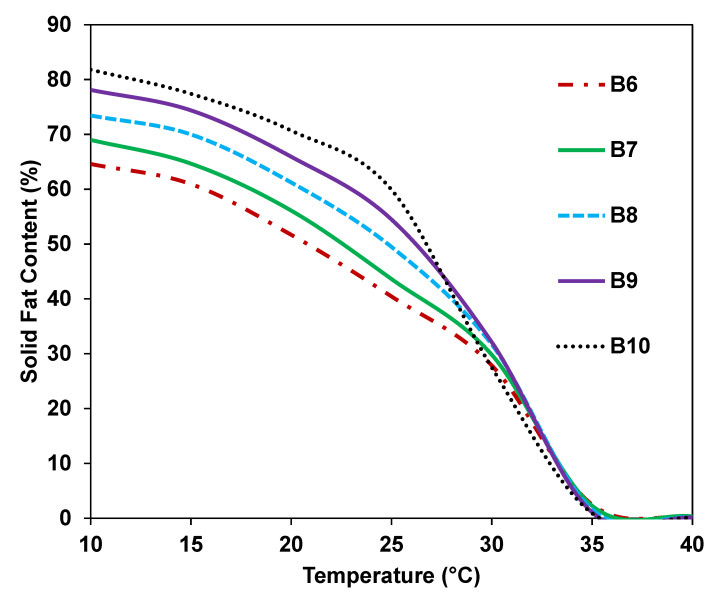
Solid fat content (%) of B6 (80% B5 and 20% CB), B7 (60% B5 and 40% CB), B8 (40% B5 and 60% CB), B9 (20% B5 and 80% CB), and B10 (0% B5 and 100% CB).

**Figure 6 foods-12-01744-f006:**
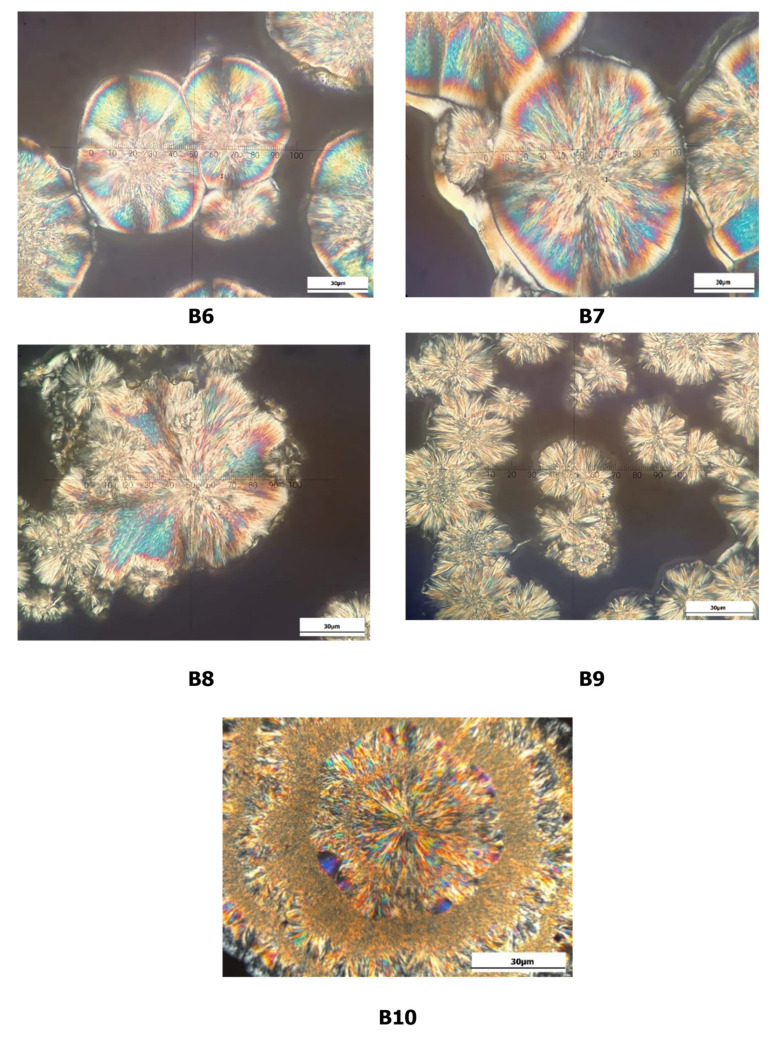
Crystal microstructure of B6 (80% B5 and 20% CB), B7 (60% B5 and 40% CB), B8 (40% B5 and 60% CB), B9 (20% B5 and 80% CB) and B10 (0% B5 and 100% CB).

**Table 1 foods-12-01744-t001:** Blending ratio for BKF-SS and POMF blends.

Blend (100 g)	BKF-SS (%)	POMF (%)
B1	90	10
B2	85	15
B3	80	20
B4	75	25
B5	70	30

**Table 2 foods-12-01744-t002:** Blending ratio for B5 and CB blends.

Blend (100 g)	B5 (%)	CB (%)
B6	80	20
B7	60	40
B8	40	60
B9	20	80
B10	0	100

**Table 3 foods-12-01744-t003:** Physicochemical properties of bambangan kernel fat stearin (BKF-SS) blended with POMF.

Physicochemical Properties	B1	B2	B3	B4	B5	POMF	CB
Iodine value (g iodine/g)	34.39 ± 0.23 ^a^	35.20 ± 0.57 ^b^	35.73 ± 0.32 ^b,c^	36.08 ± 0.01 ^c,d^	36.82 ± 0.34 ^d^	45.56 ± 0.23 ^e^	33.69 ± 0.11 ^a^
Slip melting point (°C)	34.5 ± 0.50 ^e,f^	34.25 ± 0.25 ^e^	33.67 ± 0.28 ^b,c^	33.75 ± 0.25 ^c,d^	32.67 ± 0.28 ^a^	33.50 ± 0.34 ^c,d^	33.83 ± 0.28 ^c,d^
Acid value (mg KOH/g)	2.24 ± 0.09 ^d^	2.90 ± 0.15 ^e^	1.49 ± 0.16 ^b^	1.31 ± 0.16 ^a^	1.68 ± 0.11 ^b,c^	2.33 ± 0.16 ^c^	2.81 ± 0.09 ^e^
Free fatty acid: oleic (%)	1.13 ± 0.17 ^d^	1.46 ± 0.08 ^e^	0.76 ± 0.08 ^b,c^	0.62 ± 0.08 ^a,b^	0.85 ± 0.00 ^c^	1.17 ± 0.07 ^d^	1.41 ± 0.07 ^e^
Free fatty acid: palmitic (%)	1.02 ± 0.11 ^d^	1.32 ± 0.08 ^f^	0.68 ± 0.08 ^b,c^	0.55 ± 0.08 ^a,b^	0.77 ± 0.16 ^c^	1.11 ± 0.08 ^d,e^	1.28 ± 0.11 ^e^

Values are the mean ± standard deviation of triplicate; means with a different letter within a row are significantly different (*p* < 0.05), as measured using the Tukey test. B1: 90% BKF-SS and 10% POMF, B2: 85% BKF-SS and 15% POMF, B3: 80% BKF-SS and 20% POMF, B4: 75% BKF-SS and 25% POMF, B5: 70% BKF-SS and 30% POMF.

**Table 4 foods-12-01744-t004:** Fatty acid and TAGs content (%) of BKF-SS blended with POMF.

	CB	BKF	BKF-SS	POMF	B1	B2	B3	B4	B5
Fatty acid content (%)									
C_16_	25.19 ± 0.04 ^h^	8.40 ± 0.01 ^b^	7.53 ± 0.28 ^a^	46.97 ± 0.08 ^i^	10.59 ± 0.01 ^c^	12.49 ± 0.03 ^d^	14.52 ± 0.34 ^e^	17.78 ± 0.07 ^f^	18.74 ± 0.08 ^g^
C_18_	36.40 ± 0.57 ^c^	29.16 ± 0.01 ^b^	45.86 ± 0.01 ^h^	4.77 ± 0.07 ^a^	47.89 ± 0.01 ^i^	45.39 ± 0.02 ^g^	43.21 ± 0.03 ^f^	40.26 ± 0.21 ^e^	38.26 ± 0.01 ^d^
C_18:1_	32.47 ± 0.01 ^a^	48.25 ± 0.21 ^h^	35.93 ± 0.03 ^f^	38.79 ± 0.01 ^g^	32.94 ± 0.07 ^a,b^	33.36 ± 0.21 ^c^	33.67 ± 0.47 ^d^	33.56 ± 0.02 ^c,d^	34.05 ± 0.02 ^e^
C_18:2_	2.91 ± 0.08 ^a^	9.24 ± 0.23 ^i^	4.25 ± 0.07 ^f,g^	6.83 ± 0.01 ^h^	3.29 ± 0.37 ^b^	3.52 ± 0.51 ^c^	3.72 ± 0.09 ^d^	3.96 ± 0.07 ^e^	4.14 ± 0.21 ^f^
C_18:3_	0.19 ± 0.01 ^e^	0.44 ± 0.01 ^g^	0.21 ± 0.05 ^f^	0.04 ± 0.02 ^a^	0.11 ± 0.21 ^d^	0.11 ± 0.67 ^d^	0.08 ± 0.09 ^b^	0.10 ± 0.10 ^c^	0.10 ± 0.07 ^c^
C_20_	1.09 ± 0.19 ^b^	1.75 ± 0.01 ^c^	2.20 ± 0.08 ^h^	0.38 ± 0.08 ^a^	2.29 ± 0.01 ^i^	2.11 ± 0.09 ^g^	2.09 ± 0.21 ^f^	1.96 ± 0.37 ^e^	1.87 ± 0.01 ^d^
C_22_	0.20 ± 0.00 ^b^	0.33 ± 0.01 ^e^	0.37 ± 0.11 ^f^	0.07 ± 0.21 ^a^	0.37 ± 0.09 ^f^	0.33 ± 0.10 ^e^	0.60 ± 0.07 ^g^	0.32 ± 0.01 ^d^	0.28 ± 0.01 ^c^
C_24_	0.13 ± 0.11 ^b^	0.69 ± 0.09 ^h^	0.67 ± 0.01 ^i^	0.11 ± 0.57 ^a^	0.61 ± 0.09 ^g^	0.54 ± 0.21 ^e^	0.55 ± 0.00 ^f^	0.52 ± 0.13 ^d^	0.45 ± 0.34 ^c^
TAGs composition (%)									
OLO	-	1.38 ± 0.24	0.32 ± 0.08	2.36 ± 0.02	-	1.37 ± 0.01	3.28 ± 0.11	3.98 ± 0.19	1.33 ± 0.01
POL	1.26 ± 0.07 ^c^	1.03 ± 0.01 ^b^	0.26 ± 0.01 ^a^	5.86 ± 0.11 ^i^	2.88 ± 0.34 ^d^	4.09 ± 0.09 ^g^	3.77 ± 0.57 ^e^	4.02 ± 0.02 ^f^	4.48 ± 0.09 ^h^
PLP	2.43 ± 0.43 ^c^	0.27 ± 0.09 ^b^	0.07 ± 0.13 ^a^	7.41 ± 0.07 ^i^	2.86 ± 0.01 ^d^	3.75 ± 0.01 ^g^	3.60 ± 0.21 ^f^	3.57 ± 0.37 ^e^	3.78 ± 0.67 ^h^
OOO	3.60 ± 0.01 ^h^	6.57 ± 0.01 ^i^	1.26 ± 0.13 ^a^	3.25 ± 0.09 ^e^	2.75 ± 0.43 ^c^	2.68 ± 0.17 ^b^	3.44 ± 0.19 ^g^	2.76 ± 0.01 ^d^	3.39 ± 0.00 ^f^
POO	3.42 ± 0.00 ^c^	2.68 ± 0.13 ^b^	1.16 ± 0.09 ^a^	16.36 ± 0.11 ^i^	4.08 ± 0.37 ^d^	4.61 ± 0.07 ^e^	5.02 ± 0.01 ^f^	5.14 ± 0.01 ^g^	6.53 ± 0.13 ^h^
POP	14.25 ± 0.13 ^k^	3.83 ± 0.09 ^b^	0.56 ± 0.21 ^a^	44.91 ± 0.07 ^o^	6.41 ± 0.00 ^d^	8.69 ± 0.01 ^g^	9.51 ± 0.01 ^h^	12.24 ± 0.34 ^k^	15.20 ± 0.08 ^m^
SOO	4.70 ± 0.11 ^b^	22.10 ± 0.13 ^i^	11.82 ± 0.09 ^h^	1.00 ± 0.01 ^a^	5.68 ± 0.21 ^e^	6.45 ± 0.09 ^f^	5.63 ± 0.16 ^d^	5.47 ± 0.29 ^c^	6.61 ± 0.08 ^g^
POS	33.36 ± 0.23 ^h^	10.03 ± 0.46 ^a^	15.40 ± 0.08 ^g^	10.03 ± 0.09 ^a^	11.30 ± 0.19 ^b^	11.78 ± 0.21 ^e^	11.72 ± 0.67 ^d^	10.59 ± 0.63 ^c^	11.92 ± 0.12 ^f^
SOS	25.65 ± 0.11	41.18 ± 0.09	63.87 ± 0.23	-	35.33 ± 0.58	35.85 ± 0.19	31.02 ± 0.09	30.87 ± 0.09	29.74 ± 0.01
SSS	1.62 ± 0.01	3.40 ± 0.11	2.38 ± 0.78	-	7.10 ± 0.32	6.48 ± 0.33	3.49 ± 0.21	2.07 ± 0.16	3.13 ± 0.09
Others	13.12 ± 0.01 ^e^	7.34 ± 0.17 ^b^	2.90 ± 0.17 ^a^	8.82 ± 0.68 ^c^	21.61 ± 0.30 ^i^	14.24 ± 0.08 ^f^	19.53 ± 0.13 ^h^	19.30 ± 0.20 ^g^	13.89 ± 0.07 ^d^

Values are the mean ± standard deviation of triplicate; means with a different letter within a row are significantly different (*p* < 0.05), as measured using the Tukey test. B1: 90% BKF-SS and 10% POMF, B2: 85% BKF-SS and 15% POMF, B3: 80% BKF-SS and 20% POMF, B4: 75% BKF-SS and 25% POMF, B5: 70% BKF-SS and 30% POMF, C_16_: palmitic, C_18_: stearic, C_18:1_: oleic, C_18:2_: linoleic, C_18:3_: linolenic, C_20_: arachidic, C_22_: behenic, C_24_: lignoceric, P: palmitic, S: stearic, O: oleic, L: lauric.

**Table 5 foods-12-01744-t005:** Melting properties for BKF-SS:POMF blends.

Sample	Melting Properties			
	Onset Temperature (°C)	Offset Temperature (°C)	Max Temperature (°C)	Enthalpy (J/g)
BKF	11.42 ± 0.13 ^b^	29.78 ± 0.01 ^a^	24.20 ± 0.00 ^b^	51.94 ± 0.17 ^f^
BKF-SS	22.19 ± 0.02 ^e^	37.82 ± 0.01 ^i^	31.19 ± 0.01 ^f^	66.14 ± 0.16 ^i^
POMF	11.34 ± 0.07 ^a^	30.92 ± 0.23 ^b^	22.67 ± 0.03 ^a^	53.42 ± 0.00 ^g^
CB	20.99 ± 0.58 ^c^	37.09 ± 0.08 ^g^	28.17 ± 0.11 ^c^	56.35 ± 0.12 ^h^
B1	24.51 ± 0.32 ^g^	37.22 ± 0.11 ^h^	30.32 ± 0.01 ^e^	30.61 ± 0.12 ^b^
B2	23.56 ± 0.01 ^f^	36.87 ± 0.09 ^f^	29.95 ± 0.07 ^d^	33.54 ± 0.10 ^c^
B3	25.46 ± 0.11 ^h^	35.81 ± 0.07 ^d^	32.62 ± 0.09 ^h^	34.79 ± 0.08 ^d^
B4	21.60 ± 0.43 ^d^	36.31 ± 0.67 ^e^	32.62 ± 0.07 ^h^	45.39 ± 0.09 ^e^
B5	26.71 ± 0.08 ^i^	35.26 ± 0.19 ^c^	32.11 ± 0.07 ^g^	24.30 ± 0.12 ^a^

Values are the mean ± standard deviation of triplicate; means with a different letter within a column are significantly different (*p* < 0.05), as measured using the Tukey test. B1: 90% BKF-SS and 10% POMF, B2: 85% BKF-SS and 15% POMF, B3: 80% BKF-SS and 20% POMF, B4: 75% BKF-SS and 25% POMF, B5: 70% BKF-SS and 30% POMF.

**Table 6 foods-12-01744-t006:** Fatty acid and TAG composition (%) of CBE blended with CB.

		B6	B7	B8	B9	B10
Fatty acid (%)	C_16_	21.42 ± 0.01 ^f^	22.11 ± 0.23 ^h^	23.20 ± 0.13 ^j^	24.31 ± 0.01 ^m^	25.20 ± 0.11 ^o^
	C_18_	32.92 ± 0.23 ^a^	34.02 ± 0.09 ^c,d^	34.94 ± 0.68 ^e^	36.03 ± 0.21 ^f,g^	36.40 ± 0.10 ^g^
	C_18:1_	36.80 ± 0.09 ^o^	35.88 ± 0.09 ^l^	34.75 ± 0.43 ^j^	33.88 ± 0.23 ^f^	32.47 ± 0.57 ^b^
	C_18:2_	4.63 ± 0.01 ^p^	4.20 ± 0.00 ^k^	3.79 ± 0.23 ^f^	3.38 ± 0.47 ^b^	2.91 ± 0.01 ^a^
	C_18:3_	0.14 ± 0.07 ^c^	0.16 ± 0.11 ^d^	0.17 ± 0.01 ^e^	0.18 ± 0.01 ^f^	0.18 ± 0.07 ^g^
	C_20_	1.60 ± 0.11 ^i^	1.49 ± 0.57 ^g^	1.35 ± 0.11 ^e^	1.21 ± 0.07 ^b^	1.09 ± 0.27 ^a^
	C_22_	0.28 ± 0.13 ^h^	0.26 ± 0.03 ^f^	0.24 ± 0.09 ^e^	0.22 ± 0.01 ^c^	0.20 ± 0.11 ^b^
	C_24_	0.48 ± 0.02 ^j^	0.41 ± 0.08 ^h^	0.32 ± 0.11 ^e^	0.23 ± 0.01 ^b^	0.13 ± 0.01 ^a^
TAG (%)	OLO	2.80 ± 0.21	1.09 ± 0.01	0.83 ± 0.11	-	-
	POL	4.91 ± 0.68 ^l^	2.34 ± 0.01 ^g^	1.71 ± 0.67 ^c^	1.24± 0.10 ^a^	1.26 ± 0.23 ^b^
	PLP	4.26 ± 0.23 ^j^	2.51 ± 0.67 ^e^	2.81 ± 0.56 ^g^	2.44± 0.02 ^d^	2.43 ± 0.31 ^d^
	OOO	3.90 ± 0.09 ^l^	2.66 ± 0.43 ^g^	1.92 ± 0.13 ^b^	1.61 ± 0.02 ^a^	3.60 ± 0.11 ^k^
	POO	6.85 ± 0.09 ^m^	5.52 ± 0.12 ^l^	4.51 ± 0.07 ^i^	3.89 ± 0.13 ^e^	3.42 ± 0.01 ^c^
	POP	13.58 ± 0.11 ^i^	14.75± 0.03 ^l^	14.00 ± 0.03 ^k^	13.73 ± 0.03 ^j^	14.25 ± 0.08 ^k^
	SOO	10.36 ± 0.43 ^l^	7.42 ± 0.01 ^h^	5.90 ± 0.01 ^d^	5.04 ± 0.43 ^b^	4.70 ± 0.10 ^a^
	POS	16.11 ± 0.11 ^c^	21.09 ± 0.01 ^f^	25.00 ± 0.10 ^k^	29.58 ± 0.03 ^l^	33.36 ± 0.11 ^l^
	SOS	24.96 ± 0.02 ^c^	30.17 ± 0.11 ^m^	29.94 ± 0.02 ^i^	26.89 ± 0.12 ^e^	25.65 ± 0.34 ^d^
	SSS	2.29 ± 0.01 ^c^	3.46 ± 0.23 ^h^	1.19 ± 0.02 ^a^	3.31 ± 0.23 ^e^	1.62 ± 0.03 ^b^
	Others	10.00± 0.13 ^b^	8.99 ± 0.43 ^a^	13.19 ± 0.01 ^h^	12.72 ± 0.56 ^e^	13.12 ± 0.06 ^g^

Values are the mean ± standard deviation of triplicate; means with a different letter within a row are significantly different (*p* < 0.05), as measured using the Tukey test. B6: 80% B5 and 20% CB, B7: 60% B5 and 40% CB, B8: 40% B5 and 60% CB, B9: 20% B5 and 80% CB, B10: 0% B5 and 100% CB, C_16_: palmitic, C_18_: stearic, C_18:1_: oleic, C_18:2_: linoleic, C_18:3_: linolenic, C_20_: arachidic, C_22_: behenic, C_24_: lignoceric, P: palmitic, S: stearic, O: oleic, L: lauric.

## Data Availability

Data are contained within the article or Appendix A.

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
