# Peer review of "Physical Blending of Fractionated Bambangan Kernel Fat Stearin and Palm Oil Mid-Fraction to Formulate Cocoa Butter Equivalent"

_foods, 2023, doi:10.3390/foods12091744_

Round 1
Reviewer 1 Report
This study first reported the properties of fractionated BKF-SS and POMF blends. According to their findings, the blends prepared in five ratios exhibited similar physicochemical properties to CB. Among these blends, the sample that contains 70% bambangan kernel fat stearin and 30% palm oil mid-fraction showed remarkable similarity to CB. This experiment was designed well. A major revision is needed before further consideration.
- Line 130, why do the samples need to incubate for seven days at 26℃ to stabilize and temper the fat?
- In Table 3, please use the complete name for SMP, AV. Also, I would suggest the authors include CB in this table.
- In Tables 4 & 5, B5 is significantly different from the CB, which makes the conclusion less convincing.
- In Figure 5, I would suggest name what B6, B7, B8, B9, and B10 are.
Reviewer 2 Report
The manuscript titled "Physical blending of fractionated bambangan kernel fat stearin and palm oil mid-fraction to formulate cocoa butter equivalent", aim to formulate a butter similar to cocoa butter. The authors formulated mixtures of fractionated bambangan kernel fat stearin and palm oil mid-fraction.
Some comments to improve the manuscript:
Line 269. Is there a relationship between the size of the crystals, with Tp of fusion? Explain
3.3. Melting properties. Explain the behavior of the enthalpy of melting.
Tables. All abbreviations used in tables should be defined in the table note
Reviewer 3 Report
The focus of this study is original, the experimental design is complete and the authors well described and commented the obtained results, also by using an effective combination of figures and tables. Finally, the readability is very good and the references section is complete and useful for the reader.
The most important point to improve is the discussion. Indeed, the authors have fully commented all technological aspects of their study, omitting nutritional aspects. It is well-known that nutritional evaluations are essential when dealing with parameters such as fatty acids composition. Thus, the authors should add a new section in the discussion where nutritional aspects of this novel formulation are addressed.
Other points:
Please do not use acronyms and codes in the abstract which should be self-sustained.
Please improve readability and check some keying errors at lines 43, 56-58, 83, 189, Figure 2 (caption), 429-430.
Please check and correct the character °C throughout the paper and in tables. Please also use mL in the place of ml and “fatty acids” in the place of “fatty acid”.
Please be sure that the meaning of all acronyms used in this article is properly explained.
Some references are missing in the text (n.15, 23 and 32).
Lines 86-88: Is there an uncertainty value available for these temperatures?
Line 113: A 10% sample….
Line 115: please specify the supplier of filters.
Line 119: 70:30 v/v
Line 122: what about the retention time window of identification? Please specify. The addition of some examples of chromatogram is strongly recommended.
Line 127: min
Line 141: 40 h
Line 163: what type of ANOVA? Please specify.
Line 170: …content of unsaturated…
Please delete the final dot at the end of table headings.
Table 3: please add spacing: mg_KOH/g
Line 195: low if compared to? Please improve.
Line 205: please check these values. Please improve the link to Table 4.
Tables 4-5: please add the names of fatty acids to simplify the readability. Please specify the meaning of all acronyms in a footnote.
Lines 223-224: please check the values.
Line 224: high content… what about statistics? Please be sure that all comments are supported by statistics.
Line 276: please check the value 23.56.
Line 298: 10 to 50 °C? Please verify.
Figures 2, 3 and 6: please improve the readability of “30 µm”.
Line 421: what interval? Please improve.
Please simplify the sentences at lines 444-448 and 463-465.
Line 509: Lieb et al. [46]
Please check the following references for possible comparison:
Mursalin & Yernisa (2021) Cocoa Butter Substitute Production by Mixing the Fraction of Palm Kernel Stearin with Tengkawang Fat. Proceedings of the 3rd Green Development International Conference (GDIC 2020). DOI: 10.2991/aer.k.210825.029
Calliauw et al (2005) Production of cocoa butter substitutes via two-stage static fractionation of palm kernel oil. J Am Oil Chem Soc, 82, 783–789. https://doi.org/10.1007/s11746-005-1144-8
Round 2
Reviewer 3 Report
Some minor revisions needed.
Please check lines 15-16: The blends mimic? Please verify.
Line 123: The retention time window of integration should be specified. I confirm the suggestion of adding some representative chromatograms (also as supplementary material). Please note that the integration of chromatogram the authors supplied as example is wrong, since the baseline is not tangent to the peak bases. I hope the trueness of data was not compromised by such mistake.
Please remove final dots from table headings.
Line 203: ...lower AV and FFA....
Lines 232-233: please check the values. No full match with table 4.
